# Some Aspects of the Liquid Water Thermodynamic Behavior: From The Stable to the Deep Supercooled Regime

**DOI:** 10.3390/ijms21197269

**Published:** 2020-10-01

**Authors:** Francesco Mallamace, Giuseppe Mensitieri, Domenico Mallamace, Martina Salzano de Luna, Sow-Hsin Chen

**Affiliations:** 1Department of Nuclear Science and Engineering, Massachusetts Institute of Technology, Cambridge, MA 02139, USA; sowhsin@mit.edu; 2Department of Chemical, Materials and Production Engineering, University of Naples Federico II, Piazzale Teccio 80, 80125 Napoli, Italy; mensitie@unina.it (G.M.); martina.salzanodeluna@unina.it (M.S.d.L.); 3Departments of ChiBioFarAm and MIFT-Section of Industrial Chemistry, University of Messina, CASPE-INSTM, V.le F. Stagno d’Alcontres 31, 98166 Messina, Italy; mallamaced@unime.it

**Keywords:** water, metastable supercooled regime, phase and glass transition, thermodynamical functions

## Abstract

Liquid water is considered to be a peculiar example of glass forming materials because of the possibility of giving rise to amorphous phases with different densities and of the thermodynamic anomalies that characterize its supercooled liquid phase. In the present work, literature data on the density of bulk liquid water are analyzed in a wide temperature-pressure range, also including the glass phases. A careful data analysis, which was performed on different density isobars, made in terms of thermodynamic response functions, like the thermal expansion $\alpha_{P}$ and the specific heat differences $C_{P}-C_{V}$, proves, exclusively from the experimental data, the thermodynamic consistence of the liquid-liquid transition hypothesis. The study confirms that supercooled bulk water is a mixture of two liquid “phases”, namely the high density (HDL) and the low density (LDL) liquids that characterize different regions of the water phase diagram. Furthermore, the $C_{P}-C_{V}$ isobars behaviors clearly support the existence of both a liquid–liquid transition and of a liquid–liquid critical point.

## 1. Introduction

Water is the most important substance in nature, but, at the same time, it is also one of the least understood. This is due to the many anomalies that characterize its chemical-physical properties when compared to other normal liquids [1]. The best known of these is the density maximum ($\rho_{\max}$) at 277 K [2]. Another surprising feature is the polymorphism that is evident in the crystalline [3,4,5], the glass phases [6,7,8,9], and especially for the liquid phase [10]; these too accompanied by uncovered intriguing phenomena occurring upon changes in the thermodynamic conditions, in particular, pressure.

In the liquid state, the more fascinating and studied phenomenon comes from the polymorphism and the hypothesis that water has a liquid–liquid (or second) critical point (LLCP) at deeply supercooled conditions [10]. An idea proposed to interpret different experimental observations on the system thermodynamical functions that are characterized by diverging behaviors [11,12]. The verification of such a hypothesis was with continuity, over the past twenty-eight years, the body of work of many studies. Of these researches, the vast majority come from the MD simulation [13,14], often with controversial results, but rich in profound and significant suggestions (see e.g., the recent ref. [15]). However, despite these efforts, to date no unambiguous experimental proof on the existence of this second critical point has yet been found.

However, in all different phase conditions, the water (and water-systems) datum point is the hydrogen bond (HB) interaction: a non-covalent attraction between two molecules. By combining this contribution with Coulomb repulsion between electron lone-pairs on adjacent oxygen atoms, water and water-based materials can be described [16]. All of the models used to describe water are indeed based on the HB [12], whose strength and lifetimes are temperature and pressure dependent: as temperature decreases, the HB stability increases. The HB is also characterized by a “directionality”, so that the H${}_{2}$O molecules have high instantaneous asymmetry with coordination numbers that vary from two to four (or even greater) with a tetrahedrally-coordinated structure. Such a molecular packing order obeys the Pauling Ice Rule [17], except under extremely high temperature and pressure. The resulting tetrahedron, which contain two water molecules and four identical HB bonds, determines the basic structure for the bulk liquid water and ice, despite the thermodynamic fluctuations, and, at the same time, it is the basis of its polymorphism.

A singular situation is that, for glassy water, the resulting polymorphism is entirely related to the density, i.e., there are two amorphous forms of different density, generally referred to as HDA and LDA, respectively [6,7,8]. In particular, by changing the thermodynamic conditions (essentially the pressure *P*), they can be transformed one into the other. Moreover, if heated, the LDA becomes a liquid of high viscosity at about 130 K that crystallizes as cubic ice at $T_{x}=150$ K [12]. In conditions of stability for $T<T_{m}$ water would be ice (ices Ih, Ic, II, III, IV, V, etc.). Nevertheless, liquid water can be supercooled down to homogeneous nucleation temperature $T_{h}$, and in the region between $T_{x}$ and $T_{h}$ (which is called no man’s land) water cannot stably exist in its bulk liquid form. This region can only be explored through molecular dynamics studies [13], and experimentally, by confining the liquid in cavities of mesoscopic sizes, like emulsions [18,19] or in nanostructures that are smaller than the size of nucleation centers [20] or by an ultrafast melting of amorphous substrates [21].

It should be stressed that, in the 1960s, differentexperiments (i.e., scattering, calorimetry, sound propagation, and spectroscopy), carried out on liquid bulk water at low temperatures within the stable region and inside the supercooled regime, indicated the consistency of the suggestive hypothesis of a liquid polymorphism: two liquids in the same substance [22,23,24,25,26]. A situation accepted as plausible after the discovery of HDA (and, therefore, of the poly-amorphism) occurred some years later [6]. This represents the central point in the water physical-chemistry: the notion of polymorphism suggested the correct way to understand all of the abnormalities of such a system. The existence of these low density (LDL) and high density (HDL) liquids, and their thermodynamic behavior, has been since then experimentally proven in several contributions [27,28,29,30]. Together with a discontinuity of the first order in the LDA-HDA transitions [27], it has been also observed that both the two glass forms, having small entropies, are connected to the liquid at low and high pressures, respectively. In fact, the bulk liquid water, if rapidly cooled at ambient pressure, becomes LDA without crystallizing [30,31], whereas the HDL can be obtained at high at high *P* from the HDA. The latter has a structure very similar to that of the high-*P* liquid water, whereas LDA may be considered a glass form of the low-*P* liquid water [31,32,33], as also confirmed by density data.

As is well known, the water thermodynamic response functions behaviors (e.g., the isobaric specific heat, $C_{P}$, the isothermal, $\kappa_{T}$, and adiabatic, $\kappa_{S}$, compressibilities and the expansivity, $\alpha_{P}$) reflect the system anomalies, specially in the supercooled regime, where they show a singular critical-like diverging behavior [11]. In particular, the $\kappa_{T}$, as measured at 1 bar, shows a well defined maximum, appears to diverge at $T_{L}∼228$ K, and it is well fitted with a scaling law typical of the critical phenomena. These $\kappa_{T}$ behaviors have been recently observed, with a good agreement in the absolute values, by means of X-ray experiments [34], sound propagation [35], and density data [36] as far as from MD (TIP4P/2005) studies [37,38].

In particular, $\kappa_{T}$ measured, from the density data, in the pressure region 0.1–400 MPa shows the maximum only for P<200 MPa [36]. On the other hand, on decreasing the temperature, the expansivity data, representing the entropy and volume fluctuations (〈δSδV〉), show a marked anticorrelation due to molecular clustering driven by the hydrogen-bond interactions (HB) at the origin of the liquid polymorphism and, thus, of the onset and growth of the LDL component [39,40]. These data and the observation of a first-order phase transition between the two amorphous forms, added to the results of MD simulation, confirmed the LLCP idea of a liquid-liquid transition with a critical point that is located in the supercooled water region [10].

This liquid polymorphism strongly influences the dynamic properties, such as the relaxations of the system that are directly related to its fluctuations [11,18,41]. In the supercooled regime, changes of many order of magnitude in the transport parameters and a decoupling between diffusivity and viscosity are observed on approaching the amorphous phase [21,42,43,44] and their analysis agrees with the LLCP hypothesis. However, an alternative model, which is also based on the liquid polymorphism, has been proposed: the singularity free scenario, for which the observed water polymorphic changes only resemble a genuine transition [45].

The dynamic experiments made in confined water, i.e., neutron scattering [42,43] and nuclear magnetic resonance [44], rely on the Widom line (WL) concept, relevant for the LLCP hypothesis, being the locus of the maximum correlation length [46,47]. The WL is an extension of the coexistence curve, where the response functions reach extremes (and diverge on approaching the critical point), represents the locus of specific heat maxima ($C_{p}^{max}$) emanating from the critical point, and is strongly correlated with the fragile-to-strong dynamical crossover typical of supercooled liquids [42,43,46,48,49]. Finally, simulation [39,40] and experiments [44] show that the Stokes–Einstein relation breaks down on lowering the temperature just at the WL. Whereas, FTIR data [29] indicate that, on decreasing T, the population corresponding to LDL increases while that of the HDL decreases and they cross just at this line and below the LDL dominates.

As said, despite the numerous studies on the supercooled liquid water, the presence of a liquid–liquid critical point still represents an open question. In such a context, the present study addresses the thermodynamic properties of bulk water by focusing on the P−T behaviors of some response functions, namely $\kappa_{T}\left(P,T\right)$, $\alpha_{P}\left(P,T\right)$, and the specific heat difference ΔC(P,T)($\Delta C=C_{P}-C_{V}$). These physical quantities can be obtained as derivatives of the density data ρ(P,T). The main aim of the present work is to verify the liquid–liquid polymorphism as to highlighting how the obtained response functions unequivocally support the liquid-liquid transition hypothesis, and where the LLCP it is located.

For this purpose, the liquid water densities are explored and analyzed in a wide range of temperature and pressure region, which range from those of the vapor-liquid to the ones in the deep supercooled regime, also including the values corresponding to LDA and HDA phases. We consider in our work, with special care, the main findings of a recent thermodynamic model for which, in supercooled water, the phase separation is driven by entropy upon increasing the pressure [50].

## 2. Data and Results

The numerous MD simulations studies on water with different model potentials [37] and the experiments performed on confined water have given many important suggestions on what happens in the temperature region between $T_{x}$ and $T_{h}$. Examples are the existence of a density minimum [51,52] (as predicted by Percy W. Bridgman in his seminal study of more than a century ago (1912) [3]), or the discovery of the LDA existence and liquid polymorphism, or the presence of the dynamical crossover, coincident with the Widom line, etc. [20]. Here the aim is to show that the thermodynamic functions that correspond to the bulk density data (isobars) alone are sufficient to clarify the most significant water anomalies in the supercooled region including the LLCP.

Figure 1 illustrates the entire (P−T) water phase diagram from the gas to the polymorphic glass phases. Are reported as different lines the temperatures charaterizing the system like that of: (i) the glass transition, $T_{g}\left(P\right)$ [9]; (ii) the high viscosity liquid crystallization $T_{x}\left(P\right)$ [11,12]; (iii) the homogeneous nucleation $T_{h}\left(P\right)$; (iv) the melting $T_{m}\left(P\right)$ and the (v) Widom line [43], together with (vi) the dividing line between HDA and LDA ending at the LLCP (C’). The spinodal and binodal liquid-gas (BLG) lines and the triple point (TP) are also illustrated in order to highlight the system criticality and that everything, below $T_{m}$ and the BLG, is essentially metastable, whereas below the spinodal is unstable [53]. The spinodal line is obtained, according to Speedy [53], by means of the water isochores while using the IAPWS equation of state (IAPWS EoS) [54]. In the same Figure 1 are also specifically indicated the curves corresponding to the density maxima ($T_{dm}$) [11,19] and the compressibility minima ($T_{\kappa m}$) [55], the regions of metastability (between $T_{h}\left(P\right)$ and $T_{m}\left(P\right)$) with that of the ultra-viscous liquid (UVL), and the two glass transition temperatures ($T_{gL}$ and $T_{gH}$) [9].

Bulk water density isobars, from 0.1 to 800 MPa and the temperature range 100<T<370 K (i.e., from the regions of the two amorphous LDA and HDA phases and supercooled liquid state to the region of the boiling temperature), are illustrated in Figure 2. These literature data also refer to bulk liquid water inside the metastable supercooled region $T_{h}<T_{m}$ [3,19,56,57,58,59,60,61] and to LDA and HDA [6,30,31,32]. The dashed green line represents the locus of the density maxima $\rho_{\max}\left(P,T\right)$. For comparison purposes, Figure 2 also reports, data of bulk water density at ambient pressure in other conditions: (i) ice Ic; (ii) under confinement in nanotubes [51,52]; and, (iii) the results of a MD simulation of the state equation by using the TIP4P/2005 model [37].

The reported results of MD simulations and experimental data for confined water at 0.1 MPa (mainly located inside the no-man’s land) indicate the existence of both a density minimum and a continuous density evolution from the stable and metastable (supercooled) liquid phase towards the LDA values. In addition, many interesting behaviors are noticed. It can be observed that, by increasing the pressure, the value of the density maximum, $\rho_{\max}\left(P,T\right)$, increases in its absolute value, but decreases in the *T* position, disappearing near 200 MPa. At the same time, the curvature of the isobars displays a change at about the same pressure value at which such a water abnormality disappears.

It must be stressed that, depending of the experimental conditions the reported data have different accuracy (i.e., error bars). For bulk water in the stable regime, in the range 273–325 K and for 0–100 MPa the measured densities are accurate to about 6–20 ppm ($10^{-6}$ gcm${}^{-3}$), in particular it is 20 ppm at 100 MPa [58]; and for *P* from 100 to 500 MPa the error bar increases to ∼50 ppm. For bulk supercooled water at ambient pressure, up to 239 K, the experimental error was $2\times 10^{-4}$ gcm${}^{-3}$, whereas the emulsion data, in about the same temperature regime, have an experimental error of $5\times 10^{-4}$ gcm${}^{-3}$ [61]. Finally the water density obtained in the ranges 200–275 K and 40–400 MPa by using emulsified water, the measured error was ≥$5\times 10^{-3}$ gcm${}^{-3}$ [19]. Just in this latter paper there is the first observation that the plot of specific volume against temperature shows a concave-downward curvature at pressures higher than 200 MPa.

Finally, as is customary in order to evaluate the relative data standard deviation, all of the reported liquid water isobars have been fitted, with satisfactory results and without any significant systematic deviations, to a sixth-order polynomial in temperature ($\rho_{P}\left(T\right)=\sum_{n=0}^{6}a_{n}T^{n}$) [61]. E.g., for the data at the isobar of 0.1 MPa the corresponding fit is represented in the Figure as a blue line. Furthermore, the expansivity isobars $\alpha_{P}=-{\left(\partial \ln\rho /\partial T\right)}_{P}$ are calculated as the derivative of the corresponding polynomial (Figure 3, inset). The $\kappa_{T}\left(P,T\right)$ was calculated in a similar way [36].

Moreover, on cooling, at the same pressure value (200 MPa) a crossover between two different behaviors is evident. Specifically, for pressure values that are higher than 200 MPa the water isobars, corresponding to HDL water, evolve on decreasing *T*, toward the density values of the HDA that ranges from 1.15 (at 200 MPa) to 1.33 gcm${}^{-3}$ (at 800 MPa). Whereas, for pressure values that are lower than 200 MPa, all of the isobars where water is made of LDL and HDL seem to only evolve toward the LDA values that are weakly pressure dependent (0.9<ρ<0.93 gcm${}^{-3}$ in the range from 0.1 to 200 MPa). Furthermore, when the LDL contribution is dominant, there is a large probability of observing a density minimum, as suggested by simulated and confined water. In addition, 200 MPa also represents the locus of the phase diagram, at which the melting $T_{m}$ and nucleation $T_{h}$ curves change their slope from negative to positive.

Another relevant situation evidenced by the data presened in Figure 2, which are the liquid water density, by changing pressure, is characterized by a strong discontinuity; it cannot have values in the range 0.93<ρ<1.15 gcm${}^{-3}$, not only at 135 K, where we have the amorphous phases data, but also, as confirmed by the isobar’s curvature, ongoing toward the region of the supercooled liquid phase that tends to separate upon pressurizing. This is another relevant effect of the polymorphism and the high degree of cooperativity of hydrogen bond suggests, according to ref. [50], that the water liquid–liquid phase separation and the LLCP may be driven by entropy. A further insight into this aspect can be gained by analyzing the thermodynamic response functions ${\left(\partial S/\partial P\right)}_{T}$ (with $\alpha_{P}\left(P,T\right)$) and ΔC(P,T) that are derived from these density data.

The present analysis starts from these considerations and a suggestion coming from dynamical data (NMR [18]) that points to an increase of the molecular mobility with *P* up to a maximum just at ∼200 MPa. This indicates that this value of pressure determines a breaking of the HB network, thus suppressing long-range structural correlations. This aspect is confirmed by the evolution of the $\alpha_{P}\left(P,T\right)$, which represents a direct measure of the volume-entropy cross-correlations 〈δSδV〉 ($\alpha_{P}=-{\left(\partial \ln\rho /\partial T\right)}_{P}\;$ or $\alpha_{P}=\langle \delta S\delta V\rangle /k_{B}TV$ ) [55]. Just this response function gives evidence that, in simple liquids, δS and δV fluctuations become smaller as *T* decreases and are positively correlated, whereas in water, also at ambient *P*, they become more pronounced and from T<277 K are anticorrelated.

As expected, 〈δSδV〉 is negative, below $T_{dm}$, for all of the isobars presenting a density maximum $\rho_{\max}\left(T\right)$, especially in the metastable supercooled liquid state. Conversely, for the isobars above $P^{*}∼200$ MPa, at which the $\rho_{\max}$ disappears and the density curvature no more displays a downward concavity (Figure 2), 〈δSδV〉 becomes positively correlated thus providing a structural disorder in the system. Thus, in the liquid state water, an entropy decrease accompanies a cooling process, i.e., the HB molecular order increases in size and stability. For $P>P^{*}$, instead, a pressure increase tends to suppress the HB structural correlations and, thus, the local order, eventually restoring the complete disorder characteristic of normal liquids. Looking at the inset of Figure 3, which reports the expansivity, it emerges that all of the $\alpha_{P}$ isobars cross each other at a singular temperature $T^{*}≃315\pm 5$ K, while the corresponding compressibility curves show a minimum [55]. According to NMR results [62], this temperature value has been associated to the HB network [36,55,63]. The singular temperature $T^{*}$ and pressure $P^{*}$ values that emerge from behavior of the density isobars, together with that of $\alpha_{P}$ and $\kappa_{T}$, define the conditions for the LDL merging as well as its properties as compared with the other (HDL) water component [36].

The LDL (made of HB tetrahedral networks) increases its dominance over the HDL upon cooling and, solely under such a condition, liquid water can glassify into the LDA, as indicated by the density isobars displaying a downward concavity (i.e., isobars at $P<P^{*}$ for which a $\rho_{\max}$ is present). In the opposite case, which is when $P>P^{*}$, the applied pressure breaks, or deforms, the HBs thus imposing the dominance of HDL over LD. Hence, upon cooling, liquid water can only evolve into the HDA form. This picture is in full agreement with the findings of the cited neutron diffraction experiment that proved the existence of the HDL and LDL liquid phases at T=268 K by measuring the water three site-site partial structure factors [28]. At three different pressures, the HDL- LDL relative populations ($x_{\mathrm{HDL}}$) are, in fact, the following: for P=26 MPa, $x_{\mathrm{HDL}}=40\%$; for 209 MPa, $x_{\mathrm{HDL}}=60\%$ and for 400 MPa, $x_{\mathrm{HDL}}=80\%$. It must be stressed that the experimental temperature was slightly lower than $T_{m}$ only for P=26 MPa, while at the other pressures it is well inside the domain of the stable liquid phase. All of these considerations support, in terms of the bulk water density, the liquid polymorphism and LLT hypothesis. In order to further clarify the water thermodynamics, we have considered its entropy evolution ${\left(\partial S/\partial P\right)}_{T}$ and the specific heat differences $C_{P}-C_{V}=\Delta C$ evaluated from $\alpha_{P}\left(P,T\right)$ and $\kappa_{T}\left(P,T\right)$ values, according to the thermodynamic definition: (i) ${\left(\partial S/\partial P\right)}_{T}$ from $\alpha_{P}=-{\left(\partial \ln\rho /\partial T\right)}_{P}=\left(1/V\right){\left(\partial V/\partial T\right)}_{P}$ and the Maxwell relation ${\left(\partial S/\partial P\right)}_{T}=-{\left(\partial V/\partial T\right)}_{P}$ that gives $V\alpha_{P}=-{\left(\partial S/\partial P\right)}_{T}$ (or $V_{S}\alpha_{P}=\alpha_{P}/\rho =-{\left(\partial S/\partial P\right)}_{T}$, being $V_{S}=1/\rho$ the specific volume) and (ii) $C_{P}-C_{V}=\Delta C=T\alpha_{P}^{2}/\kappa_{T}\rho$, as reported in Figure 3 and Figure 4, respectively. In the latter case, the corresponding used compressibilities are those calculated in the ref. [36] by using the same density isobars reported here in the Figure 2.

## 3. Discussion

Although the two parts of Figure 3 represent the same physics, a direct view of the entropy fluctuations derivative might better illustrate the water thermodynamics according to the LLCP hypothesis. The ${\left(\partial S/\partial P\right)}_{T}$ isobars significantly change as a function of both temperature and pressure for $T<T^{*}$ and $P<P^{*}$. The first observation is that ${\left(\partial S/\partial P\right)}_{T}$ has only negative values for $P>P^{*}$ and it is a concave function with dependence on *P* and *T*. The situation is essentially the opposite for $P<P^{*}$, where ${\left(\partial S/\partial P\right)}_{T}$ is always convex. In addition, it assumes positive values and shows a divergent behavior at a certain temperature (the lines are a guide for the eyes). The data indicate that the temperature at which the divergent behavior occurs decreases with increasing pressure (becoming steeper). This evidence is consistent with the recent idea of a liquid–liquid transition driven by entropy [50] and suggests that the critical point is located at a temperature near 200 K and for a pressure range between 180 and 200 MPa. A more precise demonstration of this may come from a quantity that can be easily calculated from the data already available, namely $\Delta C=C_{P}-C_{V}$, as reported in Figure 4. Significantly different pressure-dependent behaviors characterize the stable and supercooled regime. Such a situation can be rationalized according to the fluctuation-dissipation theorem (or Le Chatelier principle), stating that any spontaneous change in the thermodynamic functions of the system in stable condition will give rise to processes that tend to restore its equilibrium. Accordingly, to such a condition, the specific heat and the compressibility should be positive at all temperatures. This also implies that $C_{P}\geq C_{V}$ and $\kappa_{T}\geq \kappa_{S}$ for all *T* under stable conditions, the equality in these relations only occurring for T=0 or $\alpha_{P}=0.$ On the other hand, near a critical transition (as $T\to T_{C}$) must be $C_{P}\gg C_{V}$ and $\kappa_{T}\gg \kappa_{S}$.

Interestingly, Figure 4 reproduces exactly these situations. From the ΔC isobars, a different behavior can be observed in the regions that are limited by $T^{*}$ and $P^{*}$. For $T>T^{*}$, the specific heat difference is higher for low pressure values, and its value decreases in value upon cooling down to $T^{*}$ at which all of the curves collapse on a single point. Below this temperature, the situation appears to reverse, being the ΔC at high pressure moderately higher than at low pressure. In addition, as shown by the drawn lines, their steepness increases with increasing pressure. Only for the isobars at $P<P^{*}\;$, where ΔC→0 it can be observed that $\alpha_{P}\to 0$; while, in the opposite case, it is always $C_{P}>C_{V}$. In particular, ΔC values are always greater than 1 JK${}^{-1}$mol${}^{-1}$ in the interval 240<T<360 K, and for the 200, 300 and 400 MPa isobars. Conversely, in the deep supercooled regime, at T<240 K, they grow (only 200 MPa isobar displays a slightly decreases). These behaviors again reflect the water changes due to the HB clustering and the onset of the LDL. For $T>T^{*}$, the only water component is HDL and its behavior is that of a normal simple liquid. A very interesting behavior, instead, is evident from the isobars at $P<P^{*}$ and for $T<T^{*}$.

As already mentioned, the specific heat measures the entropy fluctuations ($C_{P}=\langle {\left(\delta S\right)}^{2}\rangle /k_{B}T$). In the latter region of the water phase diagram (i.e., at $P<P^{*}$ and for $T<T^{*}$, and especially in the supercooled regime) just for the HB networking, they can be considered to be almost totally conformational. Furthermore, the response functions have their extremes similarly the maximum in $C_{P}$ at the Widom line (the locus of the maximum fluctuations), as proposed by the LLCP hypothesis. At the ambient pressure (0.1 MPa), such a maximum, as proposed by theoretical study linked to the LLCP hypothesis [64], just based on the Adam–Gibbs theory [65], is located inside the water no man’s land (at ∼225 K). This is fully confirmed by differential scanning calorimetry (DSC) experiments made on confined water [66]. Full agreement with these experimental data and the confirmation that such a specific heat maximum well inside the supercooled regime is essentially configurational, was given very recently, through the Adam–Gibbs theory, by using literature self diffusion data of water measured in melted ice [67].

Unfortunately, besides these observations at 0.1 MPa, to date there are no other experimental certainties on the existence of these maxima ($C_{P}$, $\kappa_{T}$) or minima ($\alpha_{P}$) in the bulk water, but only simulations studies. However, the data of Figure 3 and Figure 4 indicate that there is a certain temperature at which a divergent behavior occurs ($C_{P}\gg C_{V}$) for each pressure from 0.1 to 180 MPa. This temperature decreases with increasing pressure down to about 200 K. Therefore, all of these evidences suggest the presence of diverging fluctuations and of a critical phenomenon (with the corresponding critical point likely located in the range 180–200 MPa and 195–200 K).

## 4. Conclusions

Starting from the evidence of the water polydispersity in the solid crystalline and amorphous forms, we have considered the occurrence of a liquid polymorphism in both the stable and metastable regions in order to demonstrate the plausibility of the idea of a liquid-liquid transition and of a liquid-liquid critical point. To this aim, the thermodynamic functions $\alpha_{P}$ and $C_{P}-C_{V}$ and the related entropic fluctuations were calculated by means of the density of bulk liquid water region of the P−T phase diagram. Even if evaluated from near the boiling point to the proximity of the LLCP, in the deep supercooled regime, the behavior of these functions allows for clarifying some open questions that are related to the physical properties of this intriguing and complex system and in particular to the tetrahedral LDL component. This latter component, while having its onset in the stable liquid phase, determines the system behaviors in all of its metastable phases from the supercooled liquid to the amorphous solid. By increasing *P*, density isobars, $\rho {\left(T\right)}_{P}$, show a curvature change from concave to convex at $P^{*}∼$ 200 MPa, i.e., where density maximum disappears. The same pressure value also represents a crossover in the density evolution, upon cooling, from the supercooled liquid phase toward the glass state. All if the isobars for $P<P^{*}$ evolve only toward the low density amorphous phase LDA, while, in the opposite case, the corresponding evolution is toward the HDA. In addition, at a fixed temperature, the LDA density shows small variations when going from 0.1 MPa to $P^{*}$, whereas the HDA is largely pressure dependent. Such behaviors of the liquid water density, linked to the polyamorphism, have unusual effects on the system thermodynamics and the related functions are hence useful to understand its characteristic properties. The water thermodynamics were explored by evaluating its response functions.

Starting from these considerations, the water response functions (the coefficient of thermal expansion, the isothermal entropy derivative as a function of the pressure, and the difference between the specific heat at constant pressure and that at constant volume) have been calculated from the density isobars from 360 K to the deep supercooled regime, near the homogeneous nucleation temperature $T_{h}$, for several pressures in the interval 0.1–400 MPa.

From the obtained $\alpha_{P}$ and $C_{P}-C_{V}$ values, the first observation is that, as suggested by the density isobars curvature (Figure 2), pressure ($P^{*}$), and temperature ($T^{*}≃315\pm 5$ K) crossovers above which water behaves like any simple liquid with $\alpha_{P}>0$ in all of the explored temperature range as a consequence of an increasingly less probable intermolecular HB clustering. All of this was regardless of the fact that $\alpha_{P}$ was the response function that has received, as compared with the other, considerably less attention despite the fact that it represents the entropy and volume cross-correlations. When $P<P^{*}$ and $T<T^{*}$, and especially inside the supercooled regime, the physical properties of the system are instead dominated by the liquid polymorphism and the competition between the HDL and LDL phases. Differently, at high temperatures and pressures, the HDL component dominates; while the LDL grows by supercooling, i.e., the HB network grows, by cooling towards $T_{h}$, in sizes and lifetimes (as demonstrated by neutron experiments [28]).

The final observation, but definitely the most important one, is represented by the diverging behavior that emerges in the supercooled regime of $C_{P}-C_{V}$, a quantity that is directly linked to entropy fluctuations δS and, thus, with the space-time correlations. This evidence highlights that the observed ΔC growth is essentially of configurational origin, accompanied by divergence in the correlation functions. Such a situation, as shown by the data presented in Figure 4, can be associated with the singularities that are typical of a critical phenomenon. Unfortunately, the available density data that are reported in Figure 2 are not enough to explore the approach to the liquid–liquid critical point in a complete way. Nonetheless, the literature data collected and their analyses performed in the present work indicate, with fair precision, where the liquid-liquid critical point is located in the P−T phase diagram.

## Figures and Tables

**Figure 1 ijms-21-07269-f001:**
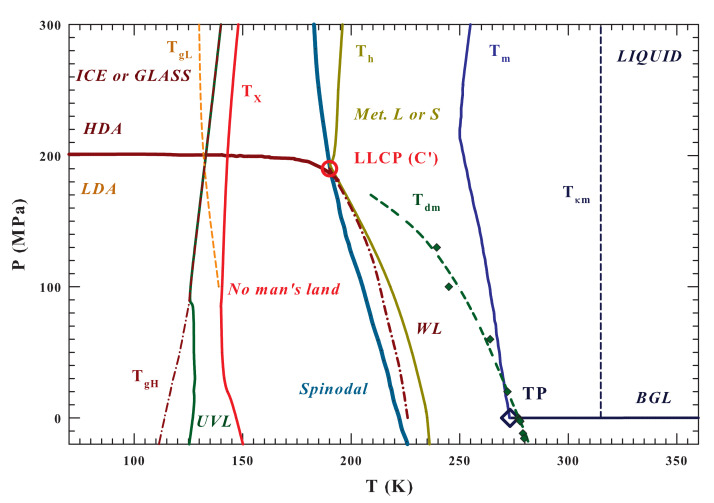
The water phase diagram in the T−P plane. The following temperature lines are illustrated: the melting $T_{M}$, the homogeneous nucleation $T_{H}$, the density maxima $T_{dm}$, the $\kappa_{T}$ minima $T_{\kappa m}$, the Widom line WL, and the crossover under stable conditions from the liquid to the crystal phase $T_{X}$ [12,20] The positions of the proposed liquid-liquid critical point $C^{\prime }$ (estimated value [11,53])) and the triple point, TP, the spinodal line (obtained from the IAPWS EoS [54]) and the binodal gas liquid (BGL) are also reported. Finally, the existence areas of the ultraviscous liquids (UVL) and the two amorphous phases LDA and HDA (together with their respective glass transition lines, $T_{gL}$ and $T_{gH}$) are proposed together with the regions characterized by the liquid polymorphysm (LDL and HDL) [9].

**Figure 2 ijms-21-07269-f002:**
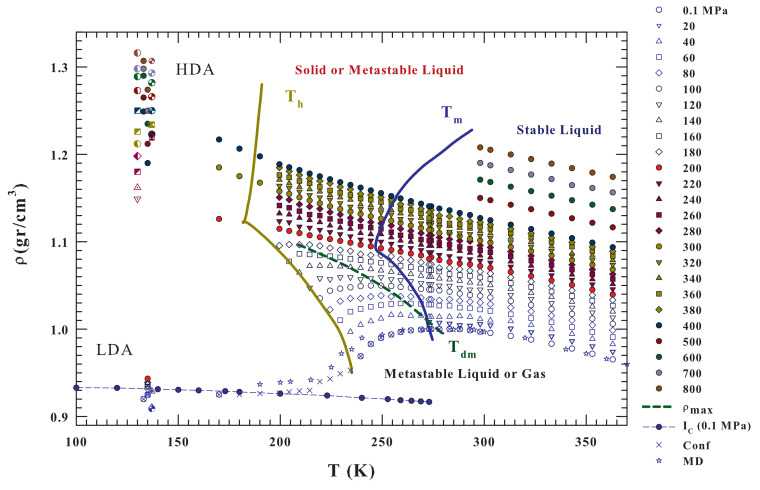
Bulk water density isobars, from 0.1 to 800MPa, in the *T* range from 100 to 370K (i.e., from the values of the two amorphous phases LDA and HDA to the stable lquid region. The corresponding data come from some different experiments; the figure also reports data of: the LDA and HDA [6,30,31,32] and bulk liquid water [3,19,56,57,58,59,60,61]. Are also included, for P=0.1 MPa, the ice Ic densites and those coming from a simulation of the equation of state by using the TIP4P/2005 water model [37] with the experimentally obtained for water inside hydrophilic nanotubes [51]. The HDA, LDA and the liquid densities, measured at a give pressure, are reported in the same color. From the data evolution inside the metastable supercooled region there is the clear suggestions that for P≥200 MPa the liquid water data evolves on to the HDA value; viceversa the lower pressures evolution is only towards the LDA.

**Figure 3 ijms-21-07269-f003:**
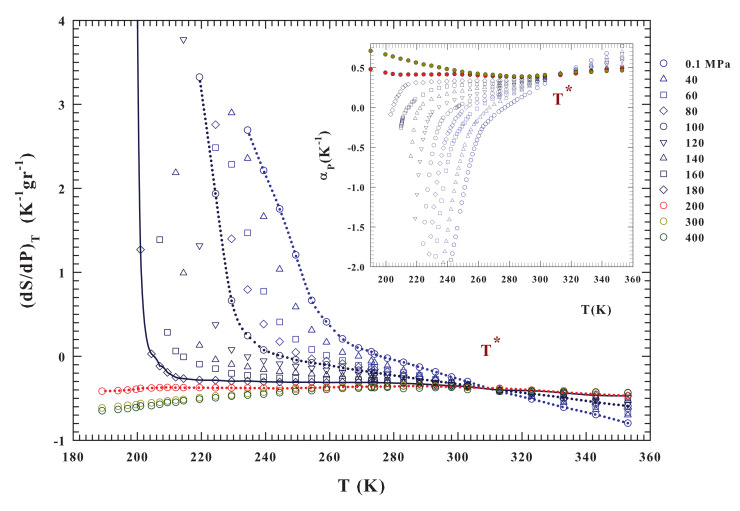
The liquid water isothermal pressure derivative of the entropy ${\left(\partial S/\partial P\right)}_{T}$ and thet thermal expansion coefficient $\alpha_{P}$ isobars (inset). These functions are calculated from the density data (Figure 2) in the range 180<T<360 K for the pressures P=0.1,40,60,80,100,120,140,160,180,200,300 and 400 MPa. For both the corresponding isobars can be observed that they cross at the same temperature $T^{*}=315\pm 5K$. In both figure and inset it is observed that the data behaviors above and below P=200 MPa are completely different in curvature and absolute values. Large entropy variations are observable only below such a pressure; while the most significant is found for P=180 MPa.

**Figure 4 ijms-21-07269-f004:**
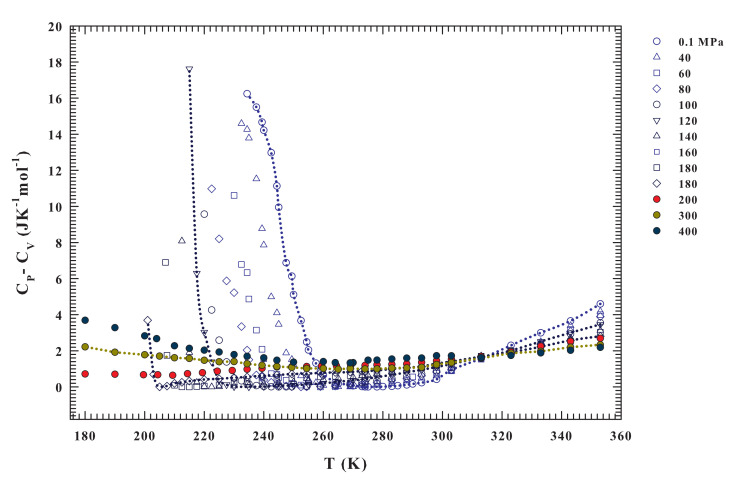
The calculated specific heat difference $C_{P}-C_{V}$ for the different pressures ranging from 0.1 to 400 MPa, illustrated in the range 180<T<360K. Also in the present case the pressure of 200 MPa is a crossover in the data behaviors; we have in fact $C_{P}\gg C_{V}$ only for P<200 MPa.

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
