# Peer review of "Some Aspects of the Liquid Water Thermodynamic Behavior: From The Stable to the Deep Supercooled Regime"

_ijms, 2020, doi:10.3390/ijms21197269_

Round 1
Reviewer 1 Report
I was looking forward to reading this manuscript on an interesting topic by a team comprising both established, well-known authorities in the field, and younger, promising scholars.
However, the level at which the manuscript is written falls well short of what I expected from the authors. It reads like an early draft, and it seems clear that it is not the product of five authors who have carefully read the manuscript. I read until the end of page 6, stopping because I feel that the referee should not put more effort into reading and correcting the paper than the authors.
The material I have read is full of sloppy mistakes, language errors, imprecise thoughts and general lack of care. I look forward to reading a significantly tightened-up manuscript.
I am attaching my version of the manuscript with some comments on the first 6 pages and on pages 15 and 16 (Figs. 1 and 2). My comments should not be taken as exhaustive.

Author Response
First of all, we would like to apologize since, by a mistake, we have originally submitted a draft and not the final revised version of our manuscript. A situation that is certainly unforgivable.
We also wish to thank all the Reviewers for their patience and the many suggestions they have provided that, for us, have been the datum point necessary to improve the manuscript quality. In this frame, we have addressed any of the points raised by the Reviewers and also added the proposed references. In particular, as suggested, we have better clarified some important issues like the Widom line and its correlations. Moreover, the manuscript has been thoroughly revised to improve cleareness and quality of English.
Reviewer 2 Report
The study by Mallamace et al., analysis bulk water density isobars over a wide range of pressure and temperature by calculating response functions. The study adds valuable input to the discussion on the existence of a metastable liquid-liquid transition at deeply supercooled conditions. What I missed when reading the manuscript, was information regarding the accuracy of the experimental data considered (i.e. error bars) and details on how the derivatives were computed, i.e. by using some interpolating function or directly from the experiment? I also expect that uncertainties in the density data are magnified if a derivative property is calculated.
Some minor note:
Why does the y-axis in Fig. 3 has no proper unit?
The manuscript should be checked for typos, e.g.
expanson -> expansion (Abstract)
represents the a P – T phase diagram (Page 2)
expotential forms (Page 5)
increases ()in its absolute value) (Page 5)
Author Response

(The authors gave the same response as above.)

Reviewer 3 Report
The paper cannot be published in the present form. It seems necessary a revision.
First of all the English is very bad. Already in the abstract there are errors, for instance: “Furthermore the behavior of the CP − CV isobars behaviors clearly support the existence of both ..”
It is not possible for a referee to correct all the errors contained in the text, so the authors must rewrite all the paper improving the English.
The introduction is rather confused. Some of the quoted papers. like 20, are interesting for historical reason but it is not clear how relevant they are in the recent debate about the liquid-liquid transition in water. The most relevant paper in the field, the ref. 31, is presented almost as a consequence of previous work, instead that paper was the starting point of all the studies about the liquid-liquid coexistence in supercooled water. Also the experiment on the LDA/HDA have been reinterpreted in terms of the possibility of the existence of a LLCP. The Widom line is not well explained and introduced in this paper. Also the connection between the Widom line and the fragile to strong crossover is not explained. Important references about the Widom line and the fragile to strong transition are missing.
The fragile to strong crossover was hypothesized by Angell
J. Phys. Chem. 97, 6339 1993 and Nature 398 1999. It was found in computer simulation for the first time by Starr et al. Phys. Rev. E 60, 6757. In experiments Xu et al. PNAS 113 2016.
The Widom line was introduced by Franzese and Stanley J. Phys Cond. Matter 19, 205126 2007
There are much better review in the literature like ref. 13 and the Chem. Rev. 116, 7463 2016. The discussion about the liquid-liquid and the LLCP is not up to date, in particular it is missing the quotation of recent interpretation like V. Holten, M.A. Anisimov, Sci. Rep. 2, 713 (2012) and the very relevant Science: Debenedetti, Sciortino and Zerze July 2020.
The fragile to strong effect was fo
It would be better for the authors to restrict the review part and concentrate on the figures and explain in details the experimental results. In any case the paper must be rewritten in a correct English.
Author Response

(The authors gave the same response as above.)

Round 2
Reviewer 3 Report
After the revision the paper can be published. Suggestion for the authors: read again the manuscript, see for instance a mistake in ref. 50
Author Response
Again, many thanks for your suggestions and comments that we have properly considered and appreciated.